# Psoriatic Arthritis: Pathogenesis and Targeted Therapies

**DOI:** 10.3390/ijms24054901

**Published:** 2023-03-03

**Authors:** Ana Belén Azuaga, Julio Ramírez, Juan D. Cañete

**Affiliations:** Rheumatology Department, Hospital Clinic and IDIBAPS of Barcelona, 08036 Barcelona, Spain

**Keywords:** psoriatic arthritis, pathogenesis, tissue heterogeneity, immune cells, synovial fibroblasts, cytokines, multiomics, single-cell RNA sequencing, biological therapy, JAK inhibitors

## Abstract

Psoriatic arthritis (PsA), a heterogeneous chronic inflammatory immune-mediated disease characterized by musculoskeletal inflammation (arthritis, enthesitis, spondylitis, and dactylitis), generally occurs in patients with psoriasis. PsA is also associated with uveitis and inflammatory bowel disease (Crohn’s disease and ulcerative colitis). To capture these manifestations as well as the associated comorbidities, and to recognize their underlining common pathogenesis, the name of psoriatic disease was coined. The pathogenesis of PsA is complex and multifaceted, with an interplay of genetic predisposition, triggering environmental factors, and activation of the innate and adaptive immune system, although autoinflammation has also been implicated. Research has identified several immune-inflammatory pathways defined by cytokines (IL-23/IL-17, TNF), leading to the development of efficacious therapeutic targets. However, heterogeneous responses to these drugs occur in different patients and in the different tissues involved, resulting in a challenge to the global management of the disease. Therefore, more translational research is necessary in order to identify new targets and improve current disease outcomes. Hopefully, this may become a reality through the integration of different omics technologies that allow better understanding of the relevant cellular and molecular players of the different tissues and manifestations of the disease. In this narrative review, we aim to provide an updated overview of the pathophysiology, including the latest findings from multiomics studies, and to describe current targeted therapies.

## 1. Introduction

Psoriasis (PsO) is a chronic inflammatory immune-mediated skin disease characterized by erythematous and scaly plaque, but which also frequently affects the scalp and nails [1]. Roughly up to 30% of patients with PsO may develop psoriatic arthritis (PsA), especially those with severe psoriasis or nail or scalp involvement [2,3]. PsA is a chronic inflammatory disease with a heterogeneous presentation involving multiple tissues and clinical domains (arthritis, spondylitis, enthesitis, and dactylitis) [4]. Persistent inflammation may lead to joint destruction and disability that might be prevented with early diagnosis and treatment [5]. Patients with PsO or PsA have more risk of develop extra-musculoskeletal inflammation and comorbidities, such as inflammatory bowel disease, uveitis and depression, cardiovascular disease, and metabolic syndrome [6,7]. Genetic studies have found risk factors associated with the transition to PsA in patients with PsO [8]. PsO and PsA share multiple genetic risk variants related to innate, adaptive immune, and autoinflammatory pathways [9]. PsA-specific genetic variants have been reported [10] and a different pattern of cytokine gene expression has been identified between skin and synovial tissue, explaining differences in the response to biologic therapies between those clinical domains [11].

The multiomics strategy is a clear step forward to precision medicine [12]. These techniques could help to increase understanding of cellular and molecular pathways to identify biomarkers for early diagnosis, prognosis, and response to treatment, as well as to discover new therapeutic targets in patients with PsA. Through the stratification of patients according to their different molecular taxonomy in the tissues involved, these technologies could help to understand the heterogeneity of the disease and lead to personalized medicine by giving the appropriate treatment to the appropriate patient. Globally, this would culminate in improving the management of the disease by reducing or preventing its disability burden and adverse effects and diminishing the economic burden [13]. 

## 2. Epidemiology

The prevalence in the general population of PsA ranges from 0.1% to 1% [14,15]. It may vary between continents, being higher in Europe than in Asia [16,17,18]. The incidence of PsA has been estimated in a meta-analysis of 83 for every 100,000 PY (95% CI, 41–167 every 100,000 person-years) [19]. Among patients with PsO, the prevalence of PsA varies according to the series. A meta-analysis estimated a prevalence of PsA in patients with PsO around 20% [3]. There are also discrepancies regarding the prevalence between men and women [20,21]. However, there are differences in the clinical manifestations depending on the sex. Studies have shown that peripheral PsA is more common in female patients, whereas axial disease is more common in male patients [22,23]. Females have less radiographic progression than male patients, but worse outcomes in pain, function, and fatigue [22,24], which contributes to poor response to treatment compared to male patients [25,26,27,28]. There is a hormonal influence in the development of the disease. Estrogens have a proinflammatory effect as they increase cytokines such as TNF, IL-6, and IL-1. In contrast, testosterone and progesterone have an anti-inflammatory effect by increasing IL-4, IL-5, and IL-10 [29]. Furthermore, the effect of testosterone levels did correlate inversely with PsA disease activity in males, but estradiol did not correlate with disease activity [30].

The presentation of PsA is usually in middle-aged people. However, up to 25% of patients may have a late onset [31]. Patients with late presentation tend to be male with a longer duration of PsO, obesity, and the presence of HLA-C*06 [32]. In addition, different studies have agreed that the appearance above 60–65 years is associated with more aggressive disease, a higher swollen joints count, higher acute phase reactants, and fatigue [33,34]. As expected, late onset is also associated with more comorbidity [34]. In addition, there are differences in terms of sex and age of presentation, male patients with early onset of PsA have greater axial involvement compared to women, while women have more family history of PsA [35].

## 3. Pathogenesis 

The pathogenesis of PsA is still far from clear, due to the heterogeneity of the tissues and pathogenic pathways involved, the disparate clinical manifestations, variable progression, and different responses to treatment. A predisposing genetic background in the presence of environmental factors, such as infections, microbiota (dysbiosis), obesity, biomechanical stress on the entheses (“deep” Koebner phenomenon) or smoking could activate the innate immune system and precipitate the onset of the disease [36]. The integration of genetic predisposition, environmental triggers, and proinflammatory cytokines is represented in Figure 1. 

### 3.1. Genetic Susceptibility

Psoriatic arthritis and PsO have a strong hereditary component. The prevalence of first-degree relatives (FDR) is around 7.7% in PsA and 17.7% in PsO [37,38]. A study using a mixed model method to assess the contribution of single nucleotide polymorphisms (SNP) from genomic wide association studies (GWAS) concluded that both PsO and PsA have a significant hereditary burden, although this is higher in PsO than in PsA [39].

The genetic region of the major histocompatibility complex (MHC) in the short arm of chromosome 6 contains several alleles or haplotypes of human leukocyte antigen (HLA) class I that are associated with an increased risk for PsO and PsA and are also associated with several clinical phenotypes of the disease. The HLA-C*06:02 association with PsO is stronger than with PsA and this allele is also associated with an early onset of PsO and a longer time between the onset of skin and joint involvement [8]. HLA-B*27, HLA-B*39, HLA-B*38, and HLA-B*08 are associated with the risk of PsA, but HLA-B*27 and HLA-B*39 are also associated with a shorter time between PsO and PsA onset. Other genotype-phenotype associations are: HLA B*08.01 with asymmetric sacroiliitis, peripheral arthritis, ankylosis and increased joint damage, whereas HLAB*27 is associated with symmetric sacroiliitis, dactylitis, and enthesitis [40]. Fine mapping of the MHC region showed that the risk heterogeneity between PsA and PsO might be driven by HLA-B amino acid at position 45, specifically glutamic acid (Glu), which is present in classical HL-B alleles associated with PsA [41]. 

In addition to the MHC complex, several SNPs in the *il23r* (IL-23 receptor), *TNFAIP3* (TNF-regulated protein A20), and *PTPN22* (tyrosine-protein phosphatase non-receptor type 22) genes, as well as an SNP within the 5q31 susceptibility locus, have a stronger association with PsA than with PsO [42]. *TNFAIP3* and *TNIP1* (TNFAIP3-interacting protein 1) encode proteins that interfere in the NF-κB pathway, resulting in negative regulation of inflammatory signaling. Therefore, variants associated with PsA could have loss of function leading to increased response to inflammatory stimulus. Variants in *IL-12B* (IL-23/IL-17 pathway), *RUNX1* (CD8-lymphocyte activation and differentiation), and *IL-13* genes have been also associated with PsA [42,43].

Deletions, insertions, and duplication events that cause copy number variants (CNVs) have been found in GWAS. A deletion in the *ADAMTS9* and *MAGI1* genes was associated with PsA, but not with PsO. *ADAMTS9* gene belongs to the family of aggrecanases, enzymes related to the cartilage extracellular matrix. The *MAGI1* gene is implicated in stabilizing adhesions and cell-to-cell contacts and probably also interferes with Tregs [10].

### 3.2. Epigenetics/DNA Methylation

Epigenetic modifications have a potential role in the disease onset, activity, response to treatment and progression in immune-mediated inflammatory diseases, as has been shown in peripheral blood monocytes in rheumatoid arthritis (RA) [44,45] and in early chronic arthritis [46]. In fact, DNA methylation may be more stable than gene expression, which is sensitive to the immediate inflammatory milieu [47]. One study showed that the differentially methylated sites in peripheral blood can distinguish PsA from PsO, suggesting that DNA methylation is a potential predictive biomarker for PsA [48]. In another study, peripheral blood CD8+ T cells from 10 PsO and 7 PsA patients concluded that patients with skin psoriasis exhibit DNA methylation patterns in CD8+ T cells that allow differentiation from PsA patients, and reflect the clinical activity of skin disease [49]. There are relatively few studies on epigenetics in PsA, generally with small size samples and it is difficult to achieve relevant clinical results. Further prospective studies are needed in order to provide insights on the role of epigenetics in search of biomarkers of the transition from PsO to PsA, the prognosis and response to therapy. 

### 3.3. Microbiome and Dysbiosis

Dysbiosis is the imbalance in the composition, distribution, or metabolic activities of the commensal species composing the normal microflora of the human body barriers [50]. The interaction between the microbiome and the immune system is responsible for many immunoregulatory mechanisms, and dysbiosis leads to alterations in barrier permeability and the consequent activation of the immune system and secretion of pro-inflammatory cytokines destabilize the junctions between epithelial cells and increase gut or skin permeability, enabling the penetration of microbes [51,52]. 

In healthy conditions, type 3 innate lymphocytes (ILC3) are implicated in gut homeostasis, as IL-23-independent IL-17A is protective in the gut by promoting epithelial cell junctions, whereas IL-23-dependent IL-17 is pathogenic [53]. In patients with spondyloarthritis (SpA), including PsA, it is suggested that ILC3 elevates the levels of IL-17A and IL-22, perpetuating inflammation, and disrupting the intestinal barrier [54]. Additionally, dysbiosis in patients with SpA can increase zonulin, increasing the permeability of the intestinal epithelium and imprinting Th9 cells to migrate to the joint [55].

Recently, a study demonstrated microbiome alterations in the skin of predominantly systemic drug-naïve PsO and PsA patients, who exhibit lower diversity compared with healthy controls, even in the absence of clinical lesions. The bacterial association network in psoriatic non-lesional (NL) skin are more similar to psoriatic lesions (L) than to healthy skin, suggesting an underlying dysbiotic process in the cutaneous surface of patients with psoriatic disease, even in the absence of clinically evident lesions. Notably, the common cutaneous commensal Corynebacterium was enriched in NL PsA compared with NL PsO, which might serve as a biomarker of disease progression [56]. These findings show that although the PsO and PsA skin microbiomes share common traits, they also exhibit differences in key taxa that might potentially be used as diagnostic biomarkers, particularly in patients with PsO at risk for disease progression to PsA [56].

### 3.4. Biomechanical Stress 

Mechanical loading is an important factor in musculoskeletal health and disease. Tendons and ligaments require physiological levels of mechanical loading to develop and maintain the tissue architecture. Pathological levels of force represent a biological (mechanical) stress that elicits an immune system-mediated tissue repair pathway in tendons and ligaments. The role of mechanical stress in ‘overuse’ injuries, such as tendinopathy, has long been known, but mechanical stress is now also emerging as a possible trigger for some forms of chronic inflammatory arthritis, including psoriatic arthritis [57]. The same authors showed that mechano-stimulation of mesenchymal cells induces CXCL1 and CCL2 for the recruitment of classical monocytes, which can differentiate into bone-resorbing osteoclasts. Therefore, biomechanical loading acts as a decisive factor in the transition from systemic autoimmunity to joint inflammation. Genetic ablation of CCL2 or pharmacologic targeting of its receptor, CCR2, abates the mechanically induced exacerbation of arthritis [58]. Thus, mechanical strain controls the site-specific localization of inflammation and tissue damage in arthritis [58]. 

The enthesis is the anatomical site where the tendons, ligaments, and joint capsule attach to the bone and it is subjected to biomechanical overload, being susceptible to mechanical stress (microtrauma) that induces the release of cytokines and growth factors, leading to secondary synovitis. Enthesitis characterizes the spondyloarthritis group, including PsA, compared with RA [59]. Interestingly, myeloid cells have been identified as the main source of local IL-23 at the enthesis [60]. Supporting the clinical relevance of these findings, PsO patients with previous bone trauma had an increased risk of PsA compared with PsO patients without trauma [61]. 

### 3.5. Obesity

In PsA patients, a study found that the odds of obesity were higher than for patients with RA, PsO, and general population [62]. Furthermore, a higher body mass index (BMI) has been associated with an increased risk of PsO and PsA [62,63]. Adipose tissue is an extremely active endocrine organ that secretes proinflammatory cytokines and cytokine-like hormones called adipokines, with either pro- or anti-inflammatory effects. Leptin is an adipokine that Is considered a link between the neuroendocrine and immune systems, with multiple effects relevant to the pathogenesis of PsA [64]. PsA and obesity share certain pathogenic mechanisms, including angiogenesis and the role of proinflammatory macrophages. On the other hand, biomechanical stress due to obesity overload could induce an aberrant response to tissue microdamage in entheseal sites, predisposing to local inflammation [64]. Taken together, these data suggest that obesity has the potential to activate many of the known immune-inflammatory pathways underlying the pathogenesis of PsA. Therefore, the link between obesity and PsA provides a potential opportunity to reduce the occurrence of PsA and improve its management by encouraging a reduction in weight, a modifiable risk factor [65]. 

### 3.6. Smoking 

Smoking has been positively associated with the risk of PsA in the general population [66], although some studies have highlighted a possible paradoxical effect of smoking in patients with psoriasis [67]. In fact, a study based on a large multinational registry of patients with a diagnosis of SpA, concluded that smoking was associated with a lower prevalence of peripheral arthritis in PsA patients [68].

### 3.7. Infections

Infections function as a trigger for the immune system to develop an immune-mediated disease. Infection by Streptococcus has been closely linked to the development of guttate PsO [69]. Even in patients with PsA, Streptococcus has been found both in synovial fluid and in peripheral blood [70]. In addition, a pharyngeal infection can increase the risk of developing PsA, as strong specific association was recently found between positive pharyngeal culture, regardless the pathogen, with an increased risk of PsA [71].

### 3.8. Innate and Adaptative Immune System: Pathological Processes in PsA

The immune-inflammatory response is triggered in patients with a genetic predisposition after interacting with certain environmental factors (dysbiosis, biomechanical stress, obesity). Locally, these factors activate the Toll-like receptors (TLRs) type 2 present in the antigen-presenting cells (APC), particularly monocytes, dendritic cells (DC), and macrophages, stimulating the release of IL-1, IL-6, TNFα, IL-17, and IL-23 through exogenous pathogen-associated molecular patterns (PAMPs) and/or endogenous damage-associated molecular patterns (DAMPs) [72,73]. Furthermore, other cells locally found in tissues may contribute to the pathophysiology, such as innate lymphoid cells (ILC), natural killer (NK) cells and mucosal associated invariant T cells. Circulating ILC type 3 (ILC3), which exhibits a Th17 profile of cytokines, are significantly increased, while ILC type 2 (ILC2) are decreased in active PsA. The ILC2/ILC3 ratio correlated negatively with disease activity in PsA (DAPSA). In addition, the presence of enthesitis, synovitis, erosions, and bone proliferation assessed by MRI and high-resolution peripheral CT (HR-pQCT) correlated with ILC3 counts in patients with PsA, suggesting the clinically relevant local effects produced by these cells [74]. 

In lymph nodes and local tissues, APC activates T lymphocytes (mainly CD8 via MHC HLA-class I), leading to the release of cytokines perpetuating the innate and adaptive type effector responses [75,76]. Naïve T lymphocytes differentiate into Th17 cells in the presence of TGFß and IL-6, inducing the expression of RORγt and leading to the production of IL-17, IL-22, IL-21, and CCL20, which favor the proliferation of Th17 [73]. Likewise, TGFß and IL-6 induce the expression of IL-23R, promoting the response to IL-23 [77], which is pathogenic [78]. IL-17, particularly IL-17A, promotes the activation of synovial fibroblasts, chondrocytes, and osteoclasts, inducing an increase in the proliferation of synovial tissue, and bone reabsorption. The thickening of the synovial tissue (ST) plus the presence of active growth factors promotes hypoxia, which induces angiogenesis (neovascularization) [78,79,80].

Even though IL-17 is key to the pathophysiology of PsA, other cytokines are needed to orchestrate the signaling pathways. IL-23 is key to the differentiation of Th17 and Th22 cells [81]. Moreover, IL-23 stimulates αβ T cells, γδ T cells, and ILC3 to produce IL-17, IL-22, TNFα, and IFN-γ, promoting typical cytokine inflammatory loops leading to persistent inflammation. On the other hand, IL-23-dependent IL-22 can induce phosphorylation of STAT3 in osteoblasts at enthesis sites, leading to bone formation [82]. In addition, IL-22 downregulates keratinocyte differentiation, inducing hyperkeratosis [83]. Finally, TNFα, IL-17, and IL-23 induce the activation of the NF-κB pathway, promoting the synthesis of more pro-inflammatory cytokines [81]. The presence of IL-12 and IFNα stimulate the Th1 response, which releases TNFα and IFN-γ and IL-2. Regulation of the inflammatory cascade requires the response mediated by Treg cells through IL-2 and TGF-ß [84]. Finally, the activation of the inflammatory cascade is reflected pathologically as dermal hyperplasia, synovitis, enthesitis, erosions, and cartilage degradation (Figure 1).

## 4. Psoriatic Synovitis 

Understanding the differences between RA and PsA synovitis is a long-sought goal, as it may be crucial in explaining the different clinical phenotypes of the two diseases and their differential response to targeted therapies. Single-cell RNA sequencing (scRNA-seq) and other omics technologies are being used to reach a deeper understanding of the cellular and molecular pathways driving the disease. We comment on classic studies in synovitis and devote a space to summarize studies using omics methodologies. 

As analyzed by immunohistochemistry, synovial inflammation in PsA is characterized by synovial membrane hypertrophy containing synovial fibroblasts (SFs) and CD68+ macrophage infiltration. Increased synovial lining hyperplasia in RA, compared with PsA, has not been definitively confirmed [85,86]. By using the Hsp47 antibody, a marker of lining and sublining SFs, we found a significant increase in sublining SFs in RA compared with PsA [87,88]. A study on the effects of the Janus kinase inhibitor (JAKi) tofacitinib (TOF) on SF function suggested that PsA SFs are activated similarly to RA SFs [89].

The sublining in PsA is similar to RA, but with a greater relative abundance of neutrophils and mast cells [90,91]. Synovial tissue macrophages are key producers of cytokines relevant to the physiopathology of chronic arthritis and their changes correlate with disease activity, radiographic progression, and response to therapy. However, a study confirmed differential p53 expression in the ST of patients with RA and PsA, as well as an association of p53 expression and CD68+ macrophages with joint damage in RA, but not in PsA [92]. Despite the pathogenic differences between RA and PsA, we found similar patterns of expression of proteins induced by GM-CSF and M-CSF in CD163+ macrophages from both diseases. Although PsA ST had significantly more CD163+ macrophages expressing the anti-inflammatory CD209 marker, the subtype predominant in ST from healthy controls, which could suggest that CD163+ CD209+ are anti-inflammatory tissue-resident macrophages, no differences were found in the expression of proinflammatory markers (activin A, TNF, MMP-12) between CD163+ macrophages CD209+ and CD209^low/−^ [93].

Dendritic cells have a key role in the initiation and progression of chronic arthritis. Recently it was identified a new DC population that derive from monocytes, characterized as CD209/CD14+ DC, which is enriched in the inflamed joint of RA and PsA, where they are further activated and exhibit a migratory phenotype, with expression of costimulatory molecules and chemokine receptors. The authors reported in this new DC subset differences in genes involved in endocytosis/antigen presentation with RA and PsA patients, which could contribute to joint inflammation. TOF specifically targets the development and functional capacity of these DCs [94].

The role of B cell in PsA synovitis remains unclear despite of the abundance of B cells and the presence of ST lymphoid neogenesis, similar to germinal center structures, has a similar frequency to RA [95]. Interestedly, serum levels of IgG autoantibodies against LL-37 and ADAMTS-L5 were correlated with the Psoriasis Area and Severity Index (PASI), and reflected disease progression in longitudinally collected serum samples from patients with psoriasis. Importantly, both anti-ADAMTS-L5 and anti-LL-37 autoantibody levels were also significantly elevated in PsO patients with PsA compared to those without PsA [96,97]. The identification of autoantibodies in PsA patients points to an autoimmune component in psoriatic disease; however, additional evidence is needed to determine the clinical utility of these autoantibodies and their contribution to disease pathogenesis [98]. B cells express surface type II MHC and co-stimulatory molecules that provide them antigen-presenting-cell features, and these phenotypic changes, although also demonstrated in psoriatic synovium, involve a larger molecule repertoire in rheumatoid synovium. This support a differential antigen-presenting B-cell phenotypes in ST of RA and PsA patients [99]. Moreover, B cells can produce multiple proinflammatory cytokines and chemokines, and regulatory B cells producing IL-10 (B10 cells) are a critical anti-inflammatory B-cell subset, which are decreased in PsA [100].

Undifferentiated arthritis (UA) patients who evolved to PsA had significantly more mast cells and lining fibroblasts compared with UA patients who evolved to RA [88]. In addition, studies have shown that PsA synovitis is characterized by macroscopic hypervascularization, resulting in higher vessel density compared with RA, showing the two diseases have different expressions of pro-angiogenic factors, with PsA exhibiting elevated levels of Ang-2 and RA having increased levels of Ang-1 [101,102]. Treatment with TNF inhibitors (TNFi) therapy in PsA synovitis reduces the expression of VEGF and its receptors VGFR1 and VGFR2, but not Ang-2 expression, leading to regression of neovessels, probably by inducing endothelial cell apoptosis [80].

### 4.1. Single Cell Analysis and Other Omics Applied to Synovitis: A New Perspective on the Role of Synovial Fibroblasts

One study reported functional and transcriptional differences between subsets of fibroblasts from RA synovial tissues using subpopulation-specific mass transcriptomics and single-cell transcriptomics. A subset of fibroblasts, characterized by the expression of podoplanin, THY1 membrane glycoprotein, and cadherin-11 proteins, but lacking CD34, located in the perivascular zone of the sublining, was tripled in RA patients compared with OA patients. These fibroblasts are proliferative, secrete proinflammatory cytokines, and exhibit an in vitro phenotype characteristic of invasive cells [103]. Another study used scRNA-seq to profile ST cells from the inflamed joints of five patients with PsA and four patients with RA. Synovial fibroblast subpopulations differed between the PsA and RA samples, with an abundance of fibroblast activating protein (FAP)+THY1+ fibroblast clusters in RA tissue and THY1-fibroblast clusters in PsA. By studying cell–cell interactions, the authors found that the synergy between T-cell-derived TGFβ and macrophage-derived IL-1b drives the metabolic dysregulation of invasive pro-inflammatory synovial fibroblasts, which are expanded in RA compared with PsA. These changes show that, although there are similarities between RA and PsA, they are different diseases with different cell subtypes and functions at the local level [104].

Besides inflammation, SFs also play an important role in the transition from joint inflammation to irreversible joint damage. A recent pre-print study shows that, after treatment with IL-17A/TNF-blocking antibodies, SFs change their phenotype from a destructive IL-6+/MMP3+THY1+ to a CD200+DKK3+ subtype, actively inducing the resolution of inflammation in PsA. This phenotypic switch can be visualized due to hitherto unexplored capacities of fibroblast subtypes with regard to receptor internalization of small molecular tracers with a high affinity to FAP. Although FAP expression levels are comparable between fibroblast subtypes in the joint, the FAP internalization rate correlates with the destructive potential of fibroblasts, and resolving fibroblasts have a lower FAP internalization rate, providing a valuable imaging tool to visualize the transition from joint damage to resolution of inflammation [105]. 

### 4.2. T-Cell Subtypes in PsA

CD8+ T cells play a key role in the pathogenesis of PsA, as expected from the association of PsA with HLA-I alleles. However, the clonality of CD8+ T cells in PsA has been confirmed only recently. Using mass cytometry, a study detected a three-fold expansion of memory CD8+ T cells in the synovial fluid (SF) compared with the peripheral blood of patients with PsA. These cells exhibited clonal expansion when evaluated by scRNA-seq. Furthermore, it was shown that these cells are active and pro-inflammatory CD8+ T cells expressing the CXCR3 receptor with its ligands (CXCL9 and CXCL10) are elevated in the SF [106]. In addition, in support of the role of adaptive immunity in the pathogenesis of PsA, another study found higher levels of IL-17+CD4- (mainly CD8+) T cells as well as of IL-17+ CD4+ T cells in SF compared with peripheral blood. However, only IL-17+CD8+ T cells correlated with disease activity and joint damage progression in patients with PsA [76]. 

A study in PsA ST reported that infiltrating CD4+ T cells expressed higher levels of IL-17A, IFN-γ, GM-CSF, and CD161, with parallel enrichment of Th1, Th17, and exTh17 T-helper subsets. These polyfunctional T cells, but not the single cytokine-producing T cell subsets, correlated with disease activity (DAPSA), suggesting that polyfunctional T cells are significantly contributing to disease. The findings of this study could inform treatment decisions and the prognosis of PsA patients [107].

## 5. Targeted Therapies

Advances in the pathogenesis of PsA have identified several inflammatory pathways driven by cytokines that have been successfully targeted with specific inhibitors. Treatment of PsA is based on non-pharmacological and pharmacological measures. Both European League Against Rheumatism (EULAR) and Group for Research and Assessment of Psoriasis and Psoriatic Arthritis (GRAPPA) treatment recommendation guidelines recommend the use of conventional synthetic disease-modifying antirheumatic drugs (csDMARDs) as first-line treatment, followed by apremilast, biological therapy, or the use of targeted synthetic DMARDs (tsDMARDs) such as Janus kinase inhibitors (JAKi) [5,108]. First, TNF inhibitors were borrowed from RA but, most recently, the discovery of the key role that the IL-23/IL-17 axis plays in PsA resulted in the development of new targeted therapies for psoriatic disease. Finally, the inhibition of JAK can inhibit/reduce the signaling effects of multiple cytokines and growth factor on targeted cells. Figure 2 describes the main immune-inflammatory pathways and their targeted therapies, which we review below.

### 5.1. Apremilast

Apremilast is an oral small molecule that inhibits phosphodiesterase 4 (PDE4). PD4 inhibition promotes an increase in intracellular cyclic AMP [109], which prevents the synthesis of proinflammatory cytokines, such as TNF [110], and elevates anti-inflammatory cytokines (IL-10) [111]. Apremilast has been shown to be effective in resolving plaque psoriasis as well as other endpoints, such as nail involvement in the ESTEEM trials [112,113]. The PALACE trials have demonstrated efficacy in PsA. Randomized controlled trials (RTC) have been developed in both naive patients and with prior exposure to both biologic therapy and csDMARDs [114,115,116,117]. All RTCs had ACR20 response at week 16 as their primary endpoint. Patients treated with both doses of apremilast (20 mg or 30 mg twice daily) achieved primary and other secondary endpoints significantly more than PBO. The most common adverse effects (AE) were diarrhea, nausea, headache, and upper respiratory tract infection. Recommendation guides suggest its use for patients with skin involvement, nail involvement, peripheral PsA, enthesitis, and dactylitis.

### 5.2. TNF Inhibitors

TNFα is a master proinflammatory cytokine produced by T1 cells which promotes the activation of myeloid cells (macrophages and DC) and their production of other proinflammatory cytokines, including TNF, thus inducing systemic and local (skin and joint) inflammation. The efficacy of 5 TNFi with indications in PsA (certolizumab pegol [CZP], infliximab [IFX], adalimumab [ADA], etanercept [ETN] and golimumab [GOL]), has been shown compared with PBO [118,119]. TNFi is the first line biological therapy recommended by the most treatment guidelines for patients with PsA, proving to be effective in all PsA domains and to inhibit or decrease radiographic progression [5,120]. A meta-analysis showed no differences in the ACR20 response between ADA, ETN, and IFX [121]. ETN consists of a human recombinant protein (TNFR2) linked to a Fc region of IgG, whereas IFX, ADA, and GOL are monoclonal antibodies, and CZP is the only TNFα antagonist utilizing the Fab’ fragment of a humanized TNFα antibody which lacks the Fc region attached to two molecules of polyethylene glycol. ETN is less immunogenic than the other TNFi and does not require concomitant administration with methotrexate to maintain its long-term effectiveness [122,123]. However, ETN is less effective compared to monoclonal antibody TNFi (IFX, ADA, GOL, CZP) in the treatment of extra-musculoskeletal inflammatory manifestations associated with PsA, such as uveitis or IBD [124,125]. These differences may be related to the different mechanisms of action between only blocking soluble TNF (ETN) or, in addition, binding to transmembrane TNFα antagonist (ADA, IFX, GOL, CZP) [126]. Due to the safety protocols and guidelines implemented after the introduction of TNFi in the clinic, the current management of these drugs is safe in the short and long term [127,128]. 

Nevertheless, like other targeted therapies, TNFi may increase the risk of infections, reactivation of latent tuberculosis (TB), and increased incidence of neoplasms. For this reason, latent TB screening is recommended, as well as discouraged its used in patients with a recent history of cancer and maintaining an up-to-date vaccination schedule. On the other hand, as a rare AE, cases of the demyelinating disease have been described [129], therefore is not recommended in people with a personal or family history of this neurological disease [130].

### 5.3. IL-17/IL-23 Axis

#### 5.3.1. IL23 Inhibitors

The IL-17/IL-23 axis is key in the development and perpetuation of PsA. The use of molecules that block these cytokines shows benefits in controlling disease activity and preventing its progression. 

Ustekinumab is a monoclonal antibody that blocks the shared p40 subunit of IL-12 and IL-23, inhibiting differentiation of Th1 and Th17 cells. Its efficacy was demonstrated for skin, peripheral arthritis, enthesitis, and dactylitis, both in RTC and in real-world data [131,132,133].

More recently, guselkumab (GSK), a monoclonal antibody that binds to the p19 subunit and specifically neutralizes IL-23, was approved after two phase 3 RCT (DISCOVER trials). In naïve patients as well as those with previous exposure to TNFi [134,135,136], the ACR 20 response at week 24 was significantly higher for patients treated with both doses of GSK than placebo. However, although each 4 week dose demonstrated significant reduction of radiographic progression compared with PBO, there were no significant differences between GSK every 8 weeks and PBO [135]. Risankizumab (RZK), a monoclonal IL-23 antibody, has been approved for the treatment of PsA. The KEEPsAKE trials in patients naïve to and with previous biological therapy showed that significantly more patients treated with RZK achieved an ACR 20 response at week 24 than PBO [137,138]. No significant differences were found in radiographic progression. Tildrakizumab (TLK) is another monoclonal IL-23 antibody. Phase 3 RCTs (INSPIRE-1 and INSPIRE-2) are currently in development [139,140], but positive results are already available from the phase 2 trial in PsA [141]. All the specific anti-IL-23 drugs have demonstrated a good safety profile, similar to previous data with ustekinumab [136], although a longer follow-up is necessary to confirm it. 

All drugs neutralizing IL-23 (GSK, RZK, and UST) have failed to show effectiveness in axial spondyloarthritis (axSpA) in RCTs [142,143]. This lack of response was unexpected, since RCTs of IL-17i have positive outcomes in the same disease [144,145]. Data from animal models of axSpA suggest that IL-23 could be required for the initiation but not the maintenance of axSpA, whereas IL-17 has a role throughout and in an IL-23-independent manner in established disease. On the other hand, a variety of innate cells, including MAIT, ILC3, iNKT, and γδ T cells, can produce IL-17 in an IL-23-independent manner [146,147], implicating them in disease pathogenesis. These observations could explain the differential response seen with IL-23 and IL-17A inhibitors (IL-23i, IL-17i) in clinical trials in axSpA [142,143,144]. The effectiveness of IL-23i in axial PsA (axPsA) has been explored in a post-hoc analysis of the DISCOVER trials, where patients with axial involvement assessed by the clinician and the presence of sacroiliitis on magnetic resonance imaging (MRI) were evaluated, suggesting that this target might be effective in axPsA [148]. Currently, an RCT with GSK in PsA patients with axial activity as its primary outcome is being developed [149] to test the hypothesis that inflammatory mechanisms are more IL-23-driven in axial PsA than in axSpA, potentially because of different proportions of IL-23R-expressing cells in the tissues. 

#### 5.3.2. IL-17 Inhibitors

Secukinumab (SEC), a fully human monoclonal antibody that selectively targets IL-17A, was approved in patients with PsA. SEC has shown superior efficacy to PBO in multiple disease domains including peripheral arthritis, spondylitis, dactylitis, enthesitis, and skin and nail disease [150,151,152,153,154]. In addition, SEC showed less radiographic progression compared to PBO. However, in ahead-to-head clinical trial, SEC was not superior to TNFi (ADA) in the musculoskeletal domain outcomes of patients with PsA (ACR 50 as primary endpoint), but was superior in the skin outcome [155]. The incidence of Candida infections in PsA is estimated to be 1.5 per 100 patient-years (PY), while the incidence of inflammatory bowel disease is 0.03–0.1 per 100 PY in PsA [156]. Ixekizumab (IXE) is also a monoclonal antibody that selectively targets IL-17A. Like SEC, IXE has been shown to be superior to PBO in ACR responses, inhibition of radiographic progression, and other clinical domains, in both TNFi refractory populations and naïve patients [157,158]. In the head-to-head clinical trial, IXE was superior to ADA in achieving the compose primary endpoint of ACR50 and a 100% improvement in the PASI 100 response at week 24, but without significant differences in the clinical musculoskeletal domains at the end of the study [159].

Bimekizumab (BMK) is a monoclonal antibody that inhibits region of IL-17 A and IL-17 F, resulting in a blockade of both homodimer and heterodimer combinations: IL-17AA, IL-17FF, and IL-17AF. Evidence from experimental and preclinical studies show that the production of IL-17A and IL-17F may be through the IL-23 classical pathway or in an IL-23-independent manner by innate and innate-like lymphocytes [79]. Furthermore, differences at the sources and signaling pathways of IL-17A and IL-17F in different stages of inflammation and through different tissues indicate that IL-17A and IL-17F contribute independently to chronic tissue inflammation, having non-redundant roles in axPsA or axSpA [160]. Therefore, dual inhibition of IL-17A and IL-17F might provide better outcomes than IL-17A blockade alone. BMK phase 3 RCT in PsA (BE COMPLETE) included patients previously exposed to TNFi, who were randomized to receive BMK 160 mg or PBO. The primary objective was the ACR 50 response at week 16, which was achieved by 43% of patients treated with BMK compared with 7% with PBO. The PASI 90 was reached at week 16 in more patients treated with BMK vs. PBO (69% vs. 7%; *p* < 0.0001) [161]. The BE OPTIMAL RCT included a third arm with an active comparator (ADA). At week 16, BMK and ADA-treated patients had a higher ACR50 response and there was evidence of less radiographic progression at week 16 in patients treated with BMK vs. PBO [162]. Pooled data from both trials showed significantly greater resolution of enthesitis and dactylitis in patients treated with BMK vs. PBO. The AEs were similar to those presented by other IL-17i, with Candida infection in 4% and serious AEs occurred in 4% of patients treated with BMK at week 24.

Brodalumab (BRD) is a human anti-interleukin-17 receptor A (IL-17RA) monoclonal antibody which blocks its binding to IL-17 (A, F, and E). It is currently approved only for PsO [163].

Interleukin-17 inhibitors are indicated for all domains of PsA involvement, except when there is associated IBD. Despite their good safety profile, IL-17i is not recommended in patients with IBD since they do not demonstrate efficacy in RCT [164,165] and the possible worsening of the disease [166].

### 5.4. JAK Inhibitors

The janus family of intracellular kinases consists of four members: tyrosine-protein kinase 2 (TYK2), JAK1, JAK2, and JAK3. These molecules interact with various members of the signal transducers and activators of transcription (STAT) family to modulate gene transcription downstream of a variety of cell surface cytokine and growth factor receptors [167].

Tofacitinib (TOF) is a small molecule that specifically inhibits JAK1 and JAK3. Phase 3 RTC demonstrated the efficacy of TOF in relation to PBO in PsA patients naïve and also in patients who had failed with TNFi [168,169]. Regarding safety, the ORAL Surveillance study found that patients with RA treated with TOF had an increased risk of cardiovascular events compared with those treated with TNFi [170]. As a result, both the European Medicines Agency (EMA) and the U.S Food and Drug Administration (FDA) have issued a statement cautioning against the use of all JAKi as the first option in patients over 65 years of age, smokers, and those with cardiovascular risk factors, a history of thromboembolic events, or a history of malignancy.

Upadacitinib (UPA) is a small molecule that inhibits JAK1. In 2021 the dose of 15 mg/day was approved for PsA. The Phase 3 trial, SELECT-PsA 1 demonstrated efficacy in the ACR20 response and safety compared with placebo. In addition, UPA 15 mg was not inferior to ADA while UPA 30 mg showed superiority over ADA, but also more severe adverse events [171]. Other secondary endpoints such as less radiographic progression, MDA, dactylitis resolution, and enthesitis were significantly higher in both UPA arms compared with PBO. Similarly, patients that were refractory or intolerant to TNFi (SELECT-PsA 2) achieved a significantly higher ACR20 response and MDA (36% and 45.44% with 15 and 30 mg/day, respectively) [172]. The most frequent AE was respiratory tract infection. The incidence of herpes zoster (HZ) was higher in both RCT compared with PBO.

Filgotinib (FIL) is another selective JAK1 inhibitor under development for the treatment of PsA. In the EQUATOR study, a phase 2 RCT, the ACR20 response at week 16 was achieved by 80% of patients treated with FIL and 33% of those treated with PBO. Only one patient treated with FIL had a herpes zoster infection [173]. Phase 3 RTC are currently under development in naïve patients and those previously exposed to biological therapy (PENGUIN 1 and 2) [174,175].

Deucravacitinib (DEU) is a small molecule that is a non-competitive, allosteric inhibitor of tyrosine kinase 2 (TYK2). The phase 2 trial included patients with prior exposure to TNFi and showed the efficacy of both doses (6 mg and 12 mg/day) of DEU compared with placebo in arthritis, resolution of enthesitis, and dactylitis. Most AEs were mild. No thromboembolic events or HZ were found [176]. Currently, a phase 3 RCT is ongoing to assess the efficacy and safety of DEU in patients who have not previously received treatment with biological DMARDs and those who have previously received TNFi treatment [177,178].

Brepocitinib (BRE) is an oral molecule with dual, TYK2 and JAK1, inhibitor action under investigation for the treatment of PsA. Results from a phase 2 RCT were presented at the ACR convergence 2021. Patients with previous use of DMARDs and those with prior use of one TNFi were included in the trial. A significantly higher proportion of patients treated with BRE 30 mg (66.7%) and 60 mg (74.6%) achieved the primary endpoint (ACR20) vs. PBO (43.4%, *p* < 0.05) at week 16. At week 16, dactylitis and enthesitis resolution were superior to placebo. HZ was found in 1.7% and no thromboembolic event was reported [179].

As mentioned above, a higher incidence of HZ has been found in patients treated with JAKi. In addition, despite being specific, when increasing the dose they can inhibit other JAK, thus increasing the possible AEs [171].

## 6. Future Perspectives in Psoriatic Arthritis

The pathophysiology of PsA is characterized by the complexity of an activated immune system with multiple cellular pathways involved, which are dynamic in the different stages or presentations of the disease and, importantly, in the distinct tissues involved. The application of single-cell techniques is making it possible to identify different cell subtypes, mainly synovial fibroblasts and T cells, with either potential pathogenic or protective roles in PsA. Hopefully, this may help to improve the development of prognostic biomarkers and future targeted therapies with new mechanisms of action. However, the clinical application of multiomics data poses major challenges, including the joint requirement for expertise and advanced facilities in statistics, biology, and computer sciences, as well as the interpretation and therapeutic actionability of molecular findings [12]. Furthermore, the heterogenous clinical features and tissues pathogenesis of PsA could pose a major challenge to disentangle the molecular taxonomy of the disease.

Although there is still much to learn about the local changes in RA and PsA synovial tissue, the progress achieved in RA by the new designs of clinical trials driven by molecular pathology in RA is impressive [180,181], and could be a model for PsA, taking into account the heterogeneity of tissues and clinical domains involved in PsA. Although the identification of biomarkers using different omics technologies (Figure 3) is still in the discovery phase, with much work to do to turn the anticipated results of these analyses into assays which are applicable in routine clinical settings [182], the EU IMI2 project HIPPOCRATES represents an exciting opportunity to address key research questions at scale and validate biomarkers for clinical implementation in PsA [183]. This will ultimately improve the quality of life for those living with PsA or at risk of developing it [131].

## 7. Conclusions

Because PsA is a complex, systemic, and heterogeneous disease in which multiple tissues and clinical domains are involved, there is a need to base disease diagnosis, classification, and management on the immunophenotypic basis of clinical domains at the tissue level, rather than clinical manifestations alone [184]. Interestingly, the experts revising unmet needs in PsA agree that “investigation of domains where tissue biopsy is not straightforward to obtain (e.g., entheses or spine), then ‘molecular imaging’ (e.g., advanced PET) with granular probes for specific cell types could help contextualize disease endotypes” [185]. In fact, we already have a successful example of the application of FAPI-PET imaging in the cellular and molecular understanding of bone damage in PsA [68].

Another area identified for improvement is the diagnosis and definition of “early PsA” or “pre-PsA” to facilitate treatment trials designed with the goal of preventing or early eradication of disease, which is ongoing with the PAMPA trial [186]. In addition, following the example of RA, new designs of molecular pathology-driven clinical trials in PsA could improve the outcome of current targeted therapies and bring us closer to the objective of personalized medicine [24].

## Figures and Tables

**Figure 1 ijms-24-04901-f001:**
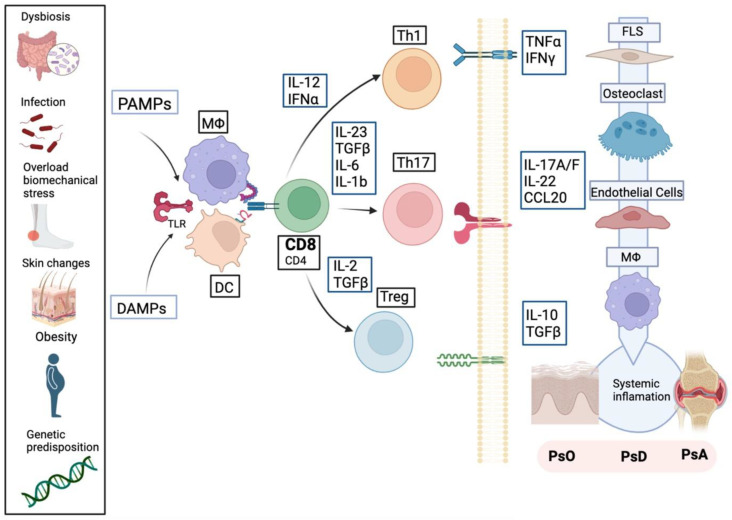
Pathological processes in psoriatic disease (PsD). Predisposing genetic background, infections, obesity, or biomechanical factors act as triggers and precipitate disease onset by activating DC macrophages which present antigens through type major histocompatibility complex (MHC) I to T cells (mainly CD8), through Toll-like receptor (TLR) type 2. This favors the local release of cytokines by triggering the innate and adaptive immune response. IL-12 and IFNα stimulate the Th1 response, which releases TNFα and IFN-γ. IL-23, TGFß, IL-6, and IL-1b activate the Th17 response in the presence of IL23, leading to the release of IL17 (mainly A isoform), IL22, IL26, and CCL20. Moreover, regulating and deactivating the inflammatory cascade requires the response mediated by Treg cells through IL-2 and TGFß. These released cytokines interact with their transmembrane receptors, promoting the release of more cytokines and attracting endothelial cells, macrophages, fibroblasts, keratinocytes, dendritic cells, epithelial cells, chondrocytes, osteoclasts, and osteoblasts. Activation of the immune system leads to synovitis, enthesitis, erosions, and lesions in the articular cartilage and skin. DAMPS (Damage-associated molecular pattern), PAMPs (Pathogen-associated molecular patterns), DC (dendritic cells), MΦ (Macrophages), CD8 (CD8 T lymphocyte), CD4 (CD4 T lymphocyte), Th1 (T helper 1 cells), Th17 (T helper 17 cells), Treg (T regulatory), FLS (synovial fibroblast).

**Figure 2 ijms-24-04901-f002:**
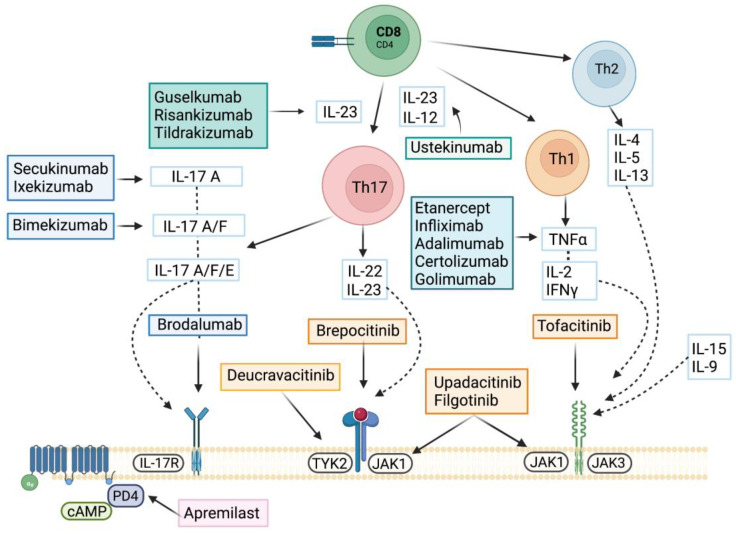
Targeted therapies. Mechanism of action of the latest approved or in-development molecules for the treatment of PsA. Interleukin 17 (IL-17) and isoforms IL-17A/F/E and IL17 receptor (IL-17R); janus kinase 1, 2, 3 (JAK1, JAK2, JAK3) and tyrosine kinase 2 (TYK2).

**Figure 3 ijms-24-04901-f003:**
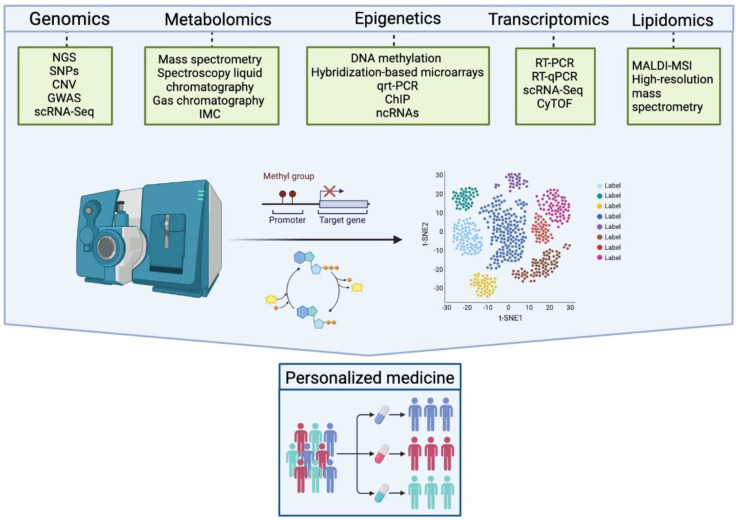
Multiomics technologies. The use of omic methods can be applied in DNA, RNA, proteins, lipids, and metabolites, among others, allowing expansion of knowledge of immune-mediated diseases to approach personalized medicine. NGS: next generation sequencing; SNP: single nucleotide polymorphisms; CNV: cellular number variations; GWAS: DNA copy number analysis; genome-wide association; scRNA-Seq: single-cell RNA Sequencing; IMC: imaging mass cytometry; qRT-PCR: quantitative reverse transcription PCR; ChIP: chromatin immunoprecipitation ; ncRNAs: histone modification and non-coding RNAs; RT-PCR: reverse transcription polymerase chain; CyTOF: single-cell with mass cytometry; MALDI-MSI: matrix-assisted laser desorption ionization mass spectrometry imaging.

## Data Availability

Not applicable.

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
