# Peer review of "Psoriatic Arthritis: Pathogenesis and Targeted Therapies"

_ijms, 2023, doi:10.3390/ijms24054901_

Round 1

Reviewer 1 Report

This is a narrative review of the pathogenesis and therapy of psoriatic arthritis. The term narrative review should appear. Although several topics are addressed the text could be much longer. In fact, each section is quite short. 
The section on environmental factors needs to be enriched.
Role of infections
Role of gender and sex hormones
A section on other cell populations such as: monocytes, APCs, and B cells (which definitely have a minor function but need to be described) should be added to the section on synovitis.
Apremilast should be added to the therapy section.
And for each drug category indication on psoriatic arthritis type and also the main contraindications.

The section on omics studies is interesting. Are there differences in this regard between the main types of psoriatic arthritis (axial vs peripheral vs enthesitis)?

Author Response

We would like to thank you for your valuable recommendations, which helped to clarify and improve the manuscript. 

Q1. This is a narrative review of the pathogenesis and therapy of psoriatic arthritis. The term narrative review should appear. Although several topics are addressed the text could be much longer. In fact, each section is quite short.

 A1. Many thanks for your suggestions. We agree this manuscript is a narrative review, and we clarify it on page 1 line 24.

Q2. The section on environmental factors needs to be enriched. Role of infections. Role of gender and sex hormones

A2. Thank you for your recommendation to complete the section on other environmental factors. We added the sex and hormonal factors in the section of epidemiology pag 2 to 3, lines 70 to 81), whereas Infections has been included on page 7, lines 244 to 249.

 Q3. A section on other cell populations such as: monocytes, APCs, and B cells (which definitely have a minor function but need to be described) should be added to the section on synovitis.

A3. Thank you. We have included a more extended view of these cells in PsA synovitis on page 9 lines 332 to 370.

 Q4. Apremilast should be added to the therapy section.

A4. We agree with the reviewer. Apremilast has been included in targeted therapies section, page 12 to 13, line 444, and lines 453 to 468, and in figure 2.

Q5. And for each drug category indication on psoriatic arthritis type and also the main contraindications.

A5. Thank you for the suggestion. We have added the corrections on page 13 to 14, lines 495 to 501; page 17, lines 595 to 597; page 18, lines 650 to 652.

Q6. The section on omics studies is interesting. Are there differences in this regard between the main types of psoriatic arthritis (axial vs peripheral vs enthesitis)?

A6. Thank you for this interesting question but, at our knowledge, currently there is not published studies on omics in the different clinical phenotypes of PsA. The few studies published until now are focused in peripheral synovitis.

Reviewer 2 Report

This is a narrative review. This review is linear and well-written, since it covers the major points related to pathogenesis and treatment of psoriatic arthritis (PsA). I have only few minor comments.

3. Psoriatic synovitis: “Although no clear autoantibodies associated with PsA have been reported, ….”. The authors are requested to consider some reports suggesting autoantibodies presence in PsA patients (e.g. DOI: 10.3389/fimmu.2018.01936).

4. Targeted therapies: Since the authors refer to targeted therapies, and not specifically to “DMARDs” (e.g. disease modifying drugs), a comment on apremilast, an oral small molecule inhibitor of phosphodiesterase 4 (PDE4) approved for PsA management, is necessary.

Author Response

We would like to thank you for your valuable recommendations, which helped to clarify and improve the manuscript. 

This is a narrative review. This review is linear and well-written, since it covers the major points related to pathogenesis and treatment of psoriatic arthritis (PsA). I have only few minor comments.

Many thanks for your positive comments.

Q1. Psoriatic synovitis: “Although no clear autoantibodies associated with PsA have been reported, ….”. The authors are requested to consider some reports suggesting autoantibodies presence in PsA patients (e.g. DOI: 10.3389/fimmu.2018.01936).

A1. We agree that is interesting to have bibliography on potential autoantibodies. We have extended the review of B-cells in PsA synovitis and we included your recommendation  at page 10, lines 356 to 360.

Q2. Targeted therapies: Since the authors refer to targeted therapies, and not specifically to “DMARDs” (e.g. disease modifying drugs), a comment on apremilast, an oral small molecule inhibitor of phosphodiesterase 4 (PDE4) approved for PsA management, is necessary.

A2. We agree with the reviewer, and we have included apremilast in targeted therapies section, page 12 to 13, line 444, and lines 453 to 468, and in figure 2.

Reviewer 3 Report

The review is a comprehensively report explaining  the integration of different omics technologies that allow  better understanding of the relevant cellular and molecular players of the different tissues involved in pscoratic inflammatory disease

1. Authors have mostly covered all the data base to comprehensively explain the review. However, to acquire reportage reporting impact authors should consider to include the epidemiology of psoriatic arthritis in different age group.

2. Data presentation seems to be outstanding

3. Minor typo errors should be fixed

4. In the conclusion/future perspective, authors can consider including hurdles or major limitations in multi omics field to properly address the disease condition

Author Response

We would like to thank you for your valuable recommendations, which helped to clarify and improve the manuscript. 

The review is a comprehensively report explaining the integration of different omics technologies that allow better understanding of the relevant cellular and molecular players of the different tissues involved in psoriatic inflammatory disease.

Many thanks for your kind and positive comments.

Q1. Authors have mostly covered all the data base to comprehensively explain the review. However, to acquire reportage reporting impact authors should consider to include the epidemiology of psoriatic arthritis in different age group.

A1. We agree with the interest of a section devoted to epidemiology in different age groups should be added. We added it in t page 2 to 3, from lines 64 to 90.

2.Data presentation seems to be outstanding

Many thanks for your kind comments.

Q3. Minor typo errors should be fixed

A3. We have revised and correct typo errors.

Q4. In the conclusion/future perspective, authors can consider including hurdles or major limitations in multi omics field to properly address the disease condition.

A4. Thank you for your interesting suggestion. This is an important part of the conclusion and future perspectives. We have added to the end of the manuscript a little sentence extracted from two experts in the field. Page 20, lines 700 to 707.

Round 2

Reviewer 1 Report

The authors addresses all the main issues. The paper is now ready to be published